# The RUNX Family, a Novel Multifaceted Guardian of the Genome

**DOI:** 10.3390/cells12020255

**Published:** 2023-01-07

**Authors:** Bibek Dutta, Motomi Osato

**Affiliations:** 1Department of Paediatrics, Yong Loo Lin School of Medicine, National University of Singapore, Singapore 117599, Singapore; 2Cancer Science Institute of Singapore, National University of Singapore, Singapore 117599, Singapore; 3International Research Center for Medical Sciences, Kumamoto University, Kumamoto 860-0811, Japan

**Keywords:** RUNX, DNA repair, retrotransposon, telomere, p53

## Abstract

The DNA repair machinery exists to protect cells from daily genetic insults by orchestrating multiple intrinsic and extrinsic factors. One such factor recently identified is the Runt-related transcription factor (RUNX) family, a group of proteins that act as a master transcriptional regulator for multiple biological functions such as embryonic development, stem cell behaviors, and oncogenesis. A significant number of studies in the past decades have delineated the involvement of RUNX proteins in DNA repair. Alterations in RUNX genes cause organ failure and predisposition to cancers, as seen in patients carrying mutations in the other well-established DNA repair genes. Herein, we review the currently existing findings and provide new insights into transcriptional and non-transcriptional multifaceted regulation of DNA repair by RUNX family proteins.

## 1. Introduction

Human cells are continuously exposed to endogenous and exogenous deadly insults that can result in severely adverse conditions such as cancer and premature aging. Cells encounter frequent endogenous damage from reactive oxygen species (ROS) and replication errors. The replication of a single human cell requires high-fidelity copying of 3 × 10^9^ bases by DNA polymerases [1]. This complicated process is not completely error-proof, and as a result, base substitutions and insertions/deletions (indels) accumulate at a rate of 10^−6^ to 10^−8^ per generation of the cell cycle [2].

Fortunately, there exists an exemplary cellular machinery that can recognize and repair diverse DNA lesions (Table 1), thereby maintaining genomic integrity. This complex evolutionarily conserved cellular mechanism is called the DNA repair machinery. The repair machinery comprises multiple different mechanisms to cope with DNA damage. This DNA repair machinery is not flawless. Sometimes the damage is beyond repair, and the cell is fated to either enter senescence or die.

This DNA repair machinery works through cohesive interaction among a multitude of cellular factors. The discoveries of such DNA repair factors were frequently initiated by their unique link to radiation-related episodes. RUNX family genes also have a tight relationship with radiation. A study of radiation-related myelodysplastic syndrome (MDS) and acute myeloid leukemia (AML) patients among Hiroshima atomic bomb survivors showed that 46% of the patients who succumbed to the disease carried point mutations in the *RUNX1* gene [3]. Moreover, US soldiers and residents who were exposed to radiation from the aboveground nuclear bomb tests in Bikini atoll or Nevada from the 1940s to 1950s developed MDS/AML harboring RUNX1 point mutations or RUNX1-related chromosomal translocation [4]. Secondly, it is well documented that RUNX1 point mutations were observed in 38% of the patients who developed secondary MDS/AML after successful treatment against primary cancers with chemotherapeutic agents with or without radiation therapy [3].

Besides the unique radiation-related phenomenon, the association with lymphocyte development, where DNA recombination is required, also suggests the potential involvement of a factor in DNA repair machinery. RUNX family proteins are well known to be involved with VDJ recombination [5,6,7]. VDJ recombination is essential for B and T lymphocyte development to be fully functional in the immune system [8,9]. VDJ recombination involves double-strand break (DSB) formation by RAG recombinases and repair through non-homologous end joining (NHEJ), and any defect in this process leads to diseases associated with compromised immunity in humans [9]. *Runx1* conditional knockout (KO) mice exhibited lymphocyte development defects due to abrogated VDJ recombination [10,11,12,13]. Moreover, direct and indirect interaction of the RUNX family with RAG and NHEJ genes has been shown [12,13,14,15,16,17].

The frequent involvement of RUNX family genes in lymphocyte development and diseases associated with radiation and chemotherapy suggests the potential roles of RUNX proteins in DNA damage repair.

## 2. RUNX Manages Reactive Oxygen Species (ROS) and Oxidative Stress

The mutations in RUNX proteins observed in patients exposed to radiation or chemotherapeutic drugs imply the importance of RUNX proteins in the downstream DNA repair pathways that deal with exogenous insults. Apart from exogenous agents, DNA damage also occurs from exposure to endogenous agents. Most of the endogenous damage arises from the interaction of DNA with reactive oxygen species (ROS) and the integration of retrotransposons [endogenous retrovirus (ERV), long interspersed class 1 elements (LINE-1)] [18,19]. RUNX proteins have been shown to directly regulate the endogenous levels of ROS and retrotransposable elements (RTEs).

ROS are endogenous agents that are by-products of the electron transport chain (ETC) [20]. These ROS form base lesions and strand breaks by damaging the methyl group and sugar residues [21,22,23]. RUNX proteins are involved in the maintenance of redox balance. ROS accumulation was hindered in leukemia-initiating cells (LIC) in T-cell acute lymphoblastic leukemia (T-ALL) through the downregulation of the protein kinase C θ (*PKC θ*) gene by NOTCH1 in a RUNX-dependent manner [24,25]. RUNX3 induced by NOTCH1 represses RUNX1, leading to the induction of PKC θ. This model postulates that an increase in RUNX1 leads to increased oxidative stress and premature senescence, but there exists another model that contradicts the role of RUNX1 in ROS manipulation. Single-cell gene expression profiling of breast acinar morphogenesis showed an inhibitor of ROS, the Forkhead transcription factor *FOXO1*, as a target gene of RUNX1 (Figure 1). This model showed that inhibition of RUNX1 and, subsequently, FOXO1 leads to an increase in overall oxidative stress [26]. Although the two models contradict each other, RUNX1 seems to play an important role in the maintenance of redox balance. Moreover, like RUNX1, RUNX3 was observed to interact with another member of the FOX gene family, FOXO3a [27]. Furthermore, in the non-small cell lung cancer model, ROS produced via ectopic expression of TGFβ was counteracted by RUNX3. RUNX3 elevates the expression of redox regulator *HMOX1*, which catalyzes the production of anti-oxidant bilirubin [28]. The other member of the family, RUNX2, may also play a role in the maintenance of ROS during osteoblastic differentiation. *RUNX2* was among the genes downregulated in hydrogen-peroxide-treated MC3T3-E1 cells [29]. Although further investigation is still needed, these findings suggest that the RUNX family takes part in the reduction of ROS, the largest endogenous genetic insult to the genome.

## 3. RUNX Regulates Retrotransposable Elements (RTEs)

Some viruses serve as exogenous DNA damage agents [30]. Retroviruses directly introduce genomic lesions as they integrate their proviruses into the host mammalian genome [31]. Notably, RUNX proteins were originally discovered as transcription factors (TFs) that bind to viral enhancers present in the regulatory region, particularly in MoMuLV [32]. Although RUNX proteins were shown to be a positive regulator for viral propagation [33,34], RUNX proteins were also shown to play a suppressive role. The unique relationship between RUNX proteins and retrovirus was recently revisited in HIV. RUNX heterodimerization factor CBFβ has been shown to bind with and stabilize the virus infectivity factor (Vif) of HIV which degrades APOBEC3 (A3). This CBFβ–Vif complex negatively impacts the transcription of RUNX-associated genes, some of which are related to DNA repair [35]. Furthermore, over-expression of RUNX1 was reported to reduce the expression of HIV-1 viral proteins and their replication [36].

The eukaryotic genome is heavily occupied (>45%) by a highly repetitive genetic component which shares similarities with retroviruses and is movable in the genome in a copy and paste manner. These elements, termed RTEs, are broadly classified into long terminal repeat (LTR) and non-log terminal repeat (non-LTR) RTEs. The expression of RTEs is involved in multiple diseases, including cancer, and can be used as a prognosis marker in AML [37]. AML with RUNX1 point mutation and inv(16) carrying CBFβ alteration showed increased RTE expression and fell into the high-risk category.

The LTR RTE family includes the ERV subfamily. ERVs constitute almost 8% of the human genome. Most of the ERV elements are non-infectious and have lost the ability to transpose in their genome due to a lack of flanking LTRs [38], but the replication and transposition of ERV can occur with the help of an autonomous retroelement, LINE-1 (L1) [39]. The regulatory regions of the ERVs include binding sites of several TFs such as RUNX1 and ETS [40]. Loss-of-function experiments on the ERV regulatory region pointed out its importance in hematopoietic development and immunity [40,41,42]. Moreover, the deregulation of ERV elements is associated with AML. Chromatin immunoprecipitation assays with sequencing (ChIP-seq) studies in AML cell lines showed clear binding enrichment of RUNX1 in ERV, and deletion of these ERV LTRs led to apoptosis in AML cell lines [43].

The other family, non-LTR RTEs, includes L1 and short interspersed elements (SINE). L1s constitute almost 17% of the human genome. L1 transcription is facilitated through a sense promoter that produces two proteins, ORF1p and ORF2p. ORF1p is involved in nuclear chaperone activity, while ORF2p provides the reverse transcription and endonuclease activity. The expression of L1 RNA and ORF proteins may occur in almost all types of cells, though the levels of expression are usually very low in the majority of somatic cells [44]. The transposability of the L1 elements makes them one of the top endogenous DNA-damaging agents [45]. Interestingly, L1 expression is modulated by the RUNX family (Figure 1). Human L1 promoter consists of RUNX3 binding sites [46,47]. Overexpression of RUNX3 leads to increased expression of L1, while RUNX1 and RUNX2 have suppressive effects [47]. Recent studies showed that irradiation increases L1 expression in hematopoietic cells, and the elevated L1 is lowered through interferon signaling mediated by exogenous thrombopoietin (TPO) [48]. RUNX1 is a transcriptional regulator of thrombopoietin receptor (MPL) and is involved with IFN-γ signaling, both of which regulate the L1 expression levels. Therefore, RUNX proteins appear to play a key role in L1 transcription. L1 insertion also leads to aberrant gene expression. One of the notable target genes of L1 insertion is *RUNX1*. A genomic insertion of L1 increased RUNX1 transcripts by (26 ± 8)-fold in human embryonic stem cells (hESCs) [49]. Considering *RUNX1* involvement in embryonic and hematopoietic development, L1-mediated abnormal *RUNX1* expression may lead to leukemia and embryonic defects. These findings support the possibility of a tight relationship between RUNX protein and L1.

## 4. RUNX Proteins Function in the Central DNA Repair Mechanism

DNA repair is an extremely complicated process. Distinct repair mechanisms are employed against different types of damage (Table 1, Figure 1). ROS- and chemical-reaction-mediated base modifications are repaired by the base excision repair pathway (BER), whereas the nucleotide excision repair pathway (NER) takes care of the nucleotide modifications imparted by UV and chemical components. Indels and base misincorporations occurring during replication errors are repaired by the mismatch repair pathway (MMR). The most dangerous form of DNA lesions, double-strand breaks (DSBs), are repaired either by the faithful homologous recombination pathway (HR) or by the error-prone NHEJ pathway. There exists another form of damage: strand crosslinks which can arise from irradiation and chemotherapeutic agents. The repair of crosslinks employs the activation of the Fanconi anemia (FA), HR, NER, and translesion synthesis (TLS) pathways.

Several cellular factors are involved in the DNA repair pathways. Amongst these proteins, a family of TFs, namely, the RUNX family, is becoming more relevant in the field of DNA repair. The RUNX family comprises RUNX1, RUNX2, and RUNX3. RUNX proteins play essential roles in several biological processes [50,51,52], and knockout of two of these proteins in mice leads to perinatal lethality [51,53,54]. The RUNX family is involved in all layers of the DNA repair machinery (Table 1, Figure 1). It has also been shown that the RUNX family exerts its DNA repair function not only via transcriptional regulation with the help of CBFβ but also through non-transcriptional regulation via physically interacting with other known DNA repair molecules.

RUNX has been shown to regulate the BER pathway. The RUNX1–ETO fusion protein, which is found in t(8:21) leukemia and functions as a dominant negative form against wild-type RUNX1, was shown to cause the downregulation of eight genes involved in BER [55,56]. Moreover, RUNX1–ETO-expressing cells were inefficient in repairing 8-oxoGuanine (8-oxoG), which is repaired by 8-oxoguanine DNA glycosylase (*OGG1*), one of the above-mentioned downregulated BER pathway genes [57,58]. These cells showed heightened sensitivity to PARP inhibitors and significant downregulation of the HR and FA pathways [59]. Further extensive studies on the role of RUNX in the BER pathway remain to be conducted.

RUNX3, a popular tumor suppressor in gastric cancer, was found to be directly associated with the sensor protein complex Ku70/80 of the NHEJ pathway through its transactivation domain (TAD) [16]. However, how this protein complex impacts the NHEJ pathway remains to be elucidated. Apart from the association with the sensor proteins, RUNX3 also directly binds with the transducer protein ATM. RUNX3 was shown to form a complex with a phosphorylated form of ATM in HeLa cells after treatment with adriamycin (ADR). It was further elucidated that RUNX3 acts as a recruiting factor of ATM onto p53 when cells are exposed to DNA damage [60]

In recent studies, RUNX proteins were shown to play a critical role in the ICL repair mediated by the FA pathway. Double knockout (DKO) mice of Runx1 and Runx3 showed co-occurrence of bone marrow failure (BMF) and myeloproliferative disorder (MPD). These phenotypes are seen in patients suffering from FA syndrome, suggesting the potential role of RUNX1 and RUNX3 in FA-pathway-mediated DNA repair [61]. Indeed, the DKO mice showed heightened sensitivity to DNA-damaging agents such as mitomycin C, which is commonly used for FA diagnosis. Mechanistically, it was further observed that both RUNX proteins physically and functionally interact with FANCI and FANCD2. Depletion of RUNX proteins significantly hampers the recruitment of FANC proteins. This recruitment of FANC proteins is independent of the RUNX/CBF-β heterodimerization-mediated transcription, suggesting non-transcriptional control on DNA repair machinery. Subsequent studies on the interaction of RUNX and FANC proteins showed that RUNX1 and RUNX3 are both poly(ADP-ribosyl)ated in a PARP-dependent manner, thereby interacting with Bloom syndrome protein (BLM) when DNA damage is introduced [62]. This interaction modulates the recruitment of FANCI/D2 in the DNA damage foci.

Damage to individual base positions also arises during replication errors, which might result in random indels and misincorporations. Dysregulation of MMR pathway genes have been associated with several cancers [63]. Two of the genes associated with the MMR pathway, *MSH6* and *SETD2,* were among the genes found to be frequently mutated in relapsed or therapy resistant ETV6/RUNX1 acute lymphoblastic leukemia (ALL) [64,65]. Mutations in MMR genes were also associated with RUNX1-mutated blast-phase chronic myeloid leukemia (BP-CML) [66,67]. Furthermore, isolated myeloid sarcoma patients (IMS) with germline *MSH6* mutation often carried a mutation in the *RUNX1* gene [68]. Apart from RUNX1, RUNX3 also seems to be involved in the MMR pathway. Knockout mouse models for *Mlh3* and *Pms2* displayed increased gastrointestinal tumor progression. Further investigation revealed Tle6-like or *TLE6D,* a member of the Transducin enhancer of Split (Tle) family, as the tumor-related amplified gene. TLE6D directly interacts with *RUNX3* and mediates the inhibition of RUNX3 transcription [69]. These studies suggest the possible involvement of RUNX proteins in MMR regulation.

## 5. RUNX Proteins May Maintain Telomere Length

Telomeres are repetitive DNA sequences of TTAGGG that form a cap and protect all the chromosomal ends [70,71]. These cap-like structures are extremely crucial for genome integrity, and telomeric dysfunction leads to devastating conditions. In the studies about RUNX proteins in FA pathways, DKO of RUNX1 and RUNX3 led to increased radiosensitivity [60], a phenotype that can result from telomere shortening [72]. Moreover, FA patients and mice deficient in the DNA repair proteins Atm, Parp, DNA-Pkcs, and Ku were shown to have altered telomere maintenance [73,74,75]. These correlations suggest a potential connection between telomere maintenance and DNA repair.

Extensive studies in yeast, SCID mice, and mammalian cells revealed the role of Ku proteins in the maintenance of telomeres [76,77,78,79]. Ku does not bind directly to the telomere DNA, but it facilitates the localization of telomeric repeat binding factors (TRF), TRF1 and TRF2, in the telomeric repeats [80,81]. Another protein of the NHEJ pathway that plays a role in telomere maintenance is DNA-PKC [82]. Inactivation of DNA-PKCs in mouse embryonic fibroblasts (MEFs) causes telomere fusions, suggesting the importance of DNA-PKCs in capping the chromosome ends. Other DNA repair proteins related to telomere maintenance are PARP-1 and ATM [73,83].

Proteins Ku, DNA-PKCs, and PARP-1 have been shown to directly interact with RUNX3. Hela-S3 cells expressing FLAG-tagged RUNX3 were subjected to SILAC (stable isotope labeling by amino acids in cell culture) to identify the RUNX3 interacting proteins. Apart from the DNA repair proteins involved in telomere maintenance, proteins involved in the cap complex and telomere maturation complex were also identified [62].

Additionally, the correlation between telomere length (TL) and mutation profile of 30 myeloid genes from 67 AML patients showed that TL is regulated by RUNX1. Patients with FLT3 and RUNX1 mutations, t(8;21), or inv(16) showed a trend towards short TL with a p-value of 0.026 [84]. Moreover, a comparison of TLs amongst patients carrying mutations in the genes related to bone marrow failure demonstrated that patients with mutations in LIG4 or RUNX1 had the shortest TLs [85]. MDS patients with RUNX1 mutations also showed a trend towards shorter TL, though it was not statistically significant [86].

Apart from its association with TL, RUNX1 has also been associated with the expression of *TERT*, one of the components of telomerase. The RUNX1-null human embryonic stem cell line GIBHe008 displayed a reduction in TERT [87]. Furthermore, RUNX1 has been observed to modulate the expression of genes such as *SIRT1* and CEBPα, which have been shown to directly regulate the expression of *TERT* [88,89,90,91]. Interestingly, recent studies have shown that TERT and TERC play an integral role in regulating the expression of RUNX2 [92,93]. These results suggest that RUNX proteins and telomere maintenance genes share an integral relationship, though further studies will be required to evaluate their in-depth relationship.

## 6. RUNX Modulates p53-Dependent Cell Death

In the scenario where DNA repair fails, one of the cell fates is apoptosis, which occurs through p53-dependent and -independent manners. RUNX family proteins are involved in modulating p53 activity either by direct interaction or through transcriptional regulation.

All three of these RUNX proteins physically interact with p53 after the induction of DNA damage by ADR, but they differ in their ways of influencing the p53 activity [60,94,95]. RUNX1 acts as a scaffold for p53–p300 binding, which facilitates the acetylation of p53 at Lys-373/382 residues [94], while RUNX3 mediates ADR-induced phosphorylation of p53 at the Ser-15 residue [60]. Unlike with RUNX1 and RUNX3, both the phosphorylation and acetylation of p53 are inhibited in the presence of RUNX2. The deacetylation of p53 was later found to be mediated by HDAC6, which acts as a binding partner of RUNX2 [95]. Apart from its role in the modification of p53, the RUNX/p53 complex was also found to transcriptionally regulate the expression of p53 downstream genes. The expression of pro-apoptotic genes BAX, NOXA, and PUMA shared a direct correlation with RUNX1 and RUNX3 expression and a negative correlation with RUNX2 expression (Figure 1).

An association between RUNX proteins and p53 has also been documented at steady state without NDA damage-inducing stress. Hematopoietic stem cells (HSCs) from *Runx1*-deficient mice displayed lower p53 protein levels but did not show a reduction in total mRNA levels, indicating the involvement of RUNX1 in the post-translational modification of p53 [96]. Similarly to Runx1 protein deficiency, loss of Runx1 methylation in HSCs also resulted in the abrogation of p53-dependent transcription and attenuation of apoptosis [97]. The transcriptional activity of RUNX1 is affected by the methylation loss, suggesting that RUNX1 also transcriptionally modulates p53. Furthermore, ChIP-seq data of RUNX1 displayed RUNX1 peaks at the p53 promoter region (data not shown). RUNX3 has also been shown to regulate p53 activity in the presence of oncogenic *RAS* expression [98]. Oncogenic RAS leads to RUNX3 activation via the MAPK pathway, and RUNX3, in turn, forms a complex with BRD2. The RUNX3/BRD2 complex induces ARF, which stabilizes p53 through the suppression of MDM2 [98]. These studies further support the notion that RUNX proteins modulate p53 through direct physical interaction and transcriptional regulation (Figure 2).

The relationship between p53 and RUNX proteins is not one-directional. p53 has also been shown to regulate RUNX levels. Lenalidomide resistance in MDS cells is conferred through mutations in or downregulation of RUNX1. Both RUNX1 mRNA and protein levels were observed to be significantly reduced in p53 KO MDS cells after lenalidomide treatment, suggesting that the downregulation of RUNX1 is induced by p53 deficiency [99]. Although RUNX proteins are primarily considered tumor suppressors, multiple studies have also shown that RUNX can also act as potential oncogenes and that p53 suppresses RUNX expression. RUNX1 inhibition often coincides with the upregulation of p53 and CBFβ in some leukemic cells. RUNX1-p53-CBFβ forms a feedback regulatory loop. RUNX1 inhibition in AML cells leads to p53 induction, which results in higher expression levels of CBFβ [100]. The elevated CBFβ stabilizes the depleted RUNX1 levels, conferring therapy resistance. Moreover, CBFβ expression was also upregulated in p53-deficient osteosarcoma, where CBFβ formed a stable complex with RUNX2 [101]. These studies suggest that p53 transcriptionally regulates RUNX proteins through CBFβ. Similar to RUNX1, RUNX3 and p53 form a regulatory axis through MDM2 expression. The induction of p53 stimulates MDM2 production in cells, which, in turn, ubiquitinates key lysine residues of RUNX3, causing proteasomal degradation [102]. Furthermore, RUNX protein levels along with MYC expression were found to be consistently upregulated in p53-deficient cancers [101,103,104]. These results suggest that p53 acts as a negative regulator for RUNX and MYC expression (Figure 2). 

## 7. RUNX Proteins Induce Senescence

Apart from cell death, failure of DNA repair leads to senescence. All three RUNX proteins have been shown to induce senescence in primary MEFs in a p53-dependent manner [59,105]. Ectopic expression of RUNX in the presence of an oncogene induces senescence via the upregulation of p19^ARF^.

RUNX1 and RUNX1–ETO both have been shown to induce senescence in primary fibroblasts and hematopoietic progenitors [59,106,107,108,109]. RUNX1 displayed senescence induction through elevated levels of p19^ARF^ expression in primary MEFs [59,109]. Similar to RUNX1, RUNX1–ETO-mediated senescence induction is dependent on p53 but independent of p19^ARF^/p14^ARF^ and p16^INK4a^ [107,109]. RUNX1–ETO is a potent inducer of ROS, activating the p38^MAPK^ pathway and elevating p53 protein levels, leading to senescence [107]. Although RUNX1–ETO expression leads to senescence, it also induces potent senescence-associated secretory phenotype (SASP), which ultimately results in escape from senescence [108]. RUNX3 acts as a regulator for ARF and p21 expression during the activation of oncogenic Ras [98]. Activation of K-RAS triggers the formation of the RUNX3–BRD2 complex, which induces the expression of ARF and p21 [98]. RUNX3 dissociates from BRD2 upon deactivation of K-RAS and forms a complex with HDAC4, which deacetylates RUNX3 and suppresses ARF and p21 expression [110]. Moreover, RUNX3 has also been reported to be a critical factor in inhibiting the progression of hepatocellular carcinoma via senescence [111]. Although RUNX2 negatively regulates the apoptotic activity of p53, it was shown that RUNX2-null MEFs escaped H-Ras^V12^-mediated senescence. This escape from senescence happened even in the presence of upregulated p38MAPK, p53, p21^Waf1^, p16^Ink4a^, and p19^Arf^. This observation suggests a role of RUNX2 downstream of the Ras/p38MAPK/p53 pathway. This phenotype was the result of elevated expression of S/G2/M cyclin genes caused by defective E2F: pRb: SWI/SNF-dependent gene repression [105,112]. These results suggest that RUNX2 may be an integral part of the SWI/SNF complex. Further studies may provide a deeper understanding of the role of RUNX proteins in senescence.

## 8. Therapeutic Applications

Mutation in the RUNX family genes causes a predisposition to cancer [113,114,115,116], at least in part due to a defective DNA repair system [117,118,119,120]. Therefore, therapeutic agents against cells carrying defective DNA pathways, such as PARP inhibitors, can be used to sensitize the cancer cells to traditional cancer therapies (chemotherapy and radiation therapy) [121,122,123,124]. Indeed, RUNX1–ETO-expressing AML cell lines were shown to be sensitive to PARP inhibitors [125,126,127]. The tight relation between *p53* gene activity and the RUNX family also opens the possibility to utilize MDM2 inhibitors [128,129,130]. MDM2, a ubiquitin ligase, functions as the principal cellular antagonist of p53 [131]. Inhibition of MDM2 will help to stabilize the impaired p53 activity in RUNX1/3-deficient cells, thus promoting apoptosis in malignant cells.

## 9. Conclusions

The evidence so far depicts the involvement of the RUNX family at multiple levels within the DNA repair process, from the regulation of stimuli to cellular outcome. Notably, RTE suppression and telomere maintenance are summarized in this review for the first time as previously unappreciated DNA-repair-related pathways mediated by RUNX. RUNX proteins perform this multi-layered regulation either through direct physical interaction or via the transcriptional regulation of genes involved with DNA repair machinery or their downstream genes. The review only summarizes the initial observations about the roles of RUNX family proteins as a new guardian of the genome. Several important questions such as those surrounding RUNX involvement in NER remain to be addressed. Further investigations need to be made to deepen our understanding about the roles of the RUNX family in DNA repair.

## Figures and Tables

**Figure 1 cells-12-00255-f001:**
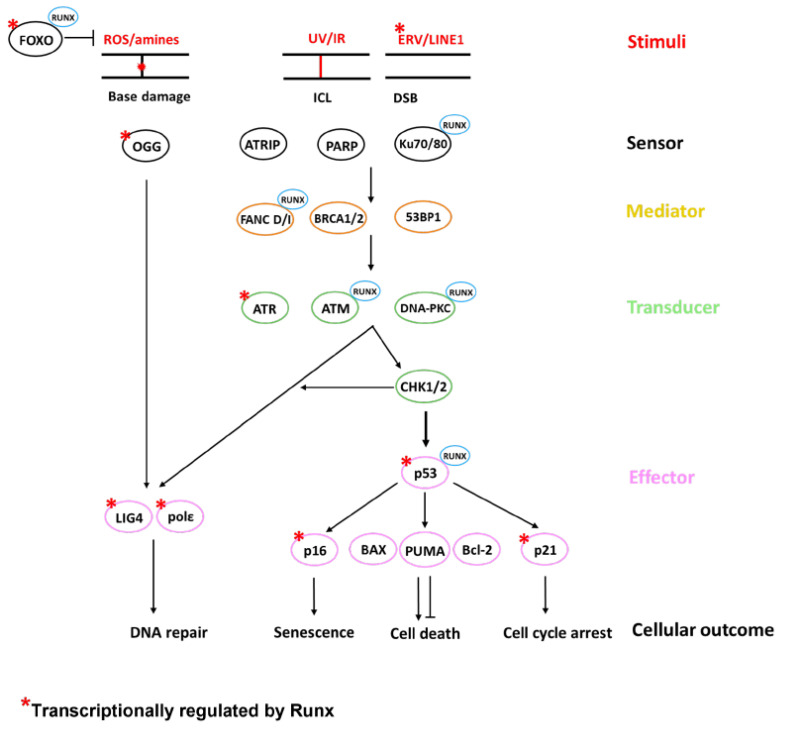
Involvement of the RUNX family in DNA repair pathways. In the scenario where a cell encounters DNA damage, the cell decides on the specific DNA repair pathway responsible for that damage. The DNA repair process is broadly divided into 5 stages: stimulus response, damage sensor, signal transducer, and mediators that provide binding of different factors and effectors. Based on the extent of the damage, the DNA might be repaired or, otherwise, the cell is fated toward apoptosis or senescence. RUNX proteins are involved in every one of these stages, from the regulation of endogenous damaging agents such as ROS and endogenous retrovirus elements (ERV/LINE-1) to the mediation of cell fate by interacting with p53 and its downstream genes. RUNX proteins regulate the DNA repair machinery either by interacting directly with the participating proteins or through transcriptional regulation. DNA repair factors that are transcriptionally regulated by RUNX1 are marked with a red asterisk. Abbreviations: ROS, reactive oxygen species; UV, ultraviolet; IR, ionizing radiation; ERV, endogenous retrovirus; ICL, inter-strand cross link; DSB, double-strand break.

**Figure 2 cells-12-00255-f002:**
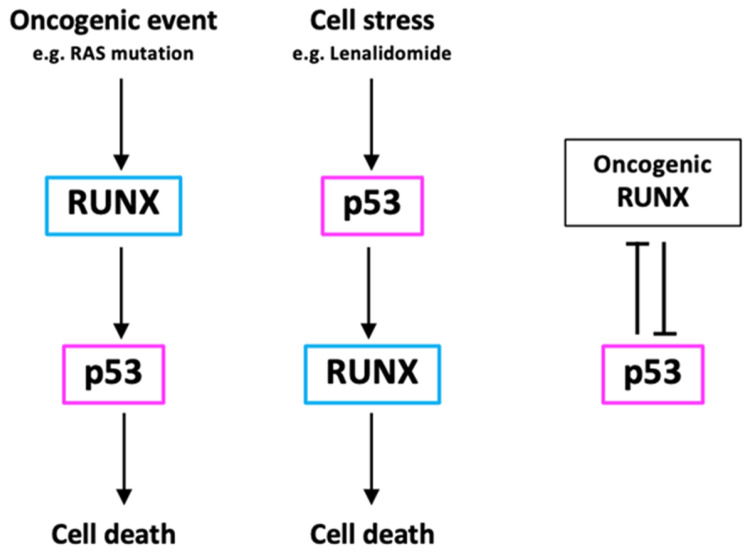
Bidirectional relationship between RUNX and p53. Wild-type RUNX and p53 positively regulate each other via transcriptional and non-transcriptional mechanisms dependent on the type of stimulation, whereas oncogenic RUNX and wild-type p53 negatively regulate one another.

**Table 1 cells-12-00255-t001:** Involvement of the RUNX family in DNA repair pathways.

Stimuli	DNA Damage Lesions	Repair Mechanism	RUNX Involvement
ROS Hydrolysis Alkylating agents Aromatic amines	Abasic sites SSBs 8-oxo-G	BER	Yes
UV Chemical agents	DNA adducts Pyrimidine dimers Glycols DNA–protein crosslink	NER	No?
IR Chemotherapeutic drugs	SSBs DSBs ICL DNA–protein crosslink	HR, FA	Yes
		NHEJ	Yes
Replication stress	Indels Base mismatch	MMR	Yes?
	Telomere erosion	TERT, TERC, ALT	Yes?
ERV, LINE-1	DSBs Indels Integration Retrotransduction	HR NHEJ MMR MMEJ	Yes?

Abbreviations: ROS, reactive oxygen species; SSB, single strand break; BER, base excision repair; NER, nucleotide excision repair; UV, ultraviolet; IR, ionizing radiation; DSB, double-strand break; ICL, inter-strand cross link; HR, homologous recombination; FA, Fanconi anemia; NHEJ, non-homologous recombination; Indel, insertion/deletion; MMR, mismatch repair; ALT, alternative lengthening of telomeres; ERV, endogenous retrovirus; MMEJ, microhomology-mediated end joining.

## Data Availability

Not applicable.

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
