# Peer review of "The RUNX Family, a Novel Multifaceted Guardian of the Genome"

_cells, 2023, doi:10.3390/cells12020255_

Round 1
Reviewer 1 Report
In this review article, the authors synthesize accumulating knowledge on the role of RUNX family transcription factors, well-known master regulators of embryonic development and cancer, on maintenance of genomic stability. They highlight both transcriptional and non-transcriptional mechanisms ranging from reducing DNA damage stimuli to regulating DNA damage repair and telomere length, to modulating cell death and senescence, to controlling retrotransposable elements (RTE). Although some of this material has been covered in other reviews, this article provides fresh perspective and importantly adds information about the role of RUNX family proteins in telomere maintenance and RTE control. However, a few minor issues should be addressed to clarify and improve the presentation.
Minor Comments:
1. The study of RTE is fast growing and it seems timely and interesting to discuss RUNX implication in RTE regulation. RTEs serve not only as DNA damage agents but also as important transcriptional regulators with oncogenic potential. Therefore, it would be reasonable to introduce this part with a new subtitle separate from the section “RUNX reduces DNA damaging stimuli”.
2. Table 1 is mentioned in the text, but is not included on the version to which the reviewer had access. This should be included in the final manuscript.
3. The statement that RUNX1 1 conditional KO mice exhibited lymphocyte developmental defects due to abrogated VDJ recombination as well as interaction of RUNX factors with RAG and NHEJ genes (page 2, 1st paragraph) need citations and References.
4. More and recent references should be cited for “RUNX proteins induce senescence”. RUNX-ETO fusion protein has been shown to induce senescence.
5. There is a typo at the end of para 4 on page 4, “ An genic insertion of L1 increased all RUNX1 transcripts by 26 ± 8 folds in the hESCs [23]”. This should be corrected.
6. There are some English grammatical errors throughout. These should be corrected.
Reviewer 2 Report
This review by Dutta and Osato describes the current knowledge regarding the functions of the RUNX family of proteins in the context of DNA repair. Although interesting, I suggest the following revisions to improve clarity and completeness:
-In multiple instances, the text lacks proper references to original research work. For example in the section "RUNX reduces DNA damaging stimuli", many of the statements made by the authors are not supported with proper references. Similarly, the section "Therapeutic applications" is devoided of any reference. The main interest of a review is to present the current knowledge on a given topic but this requires to fully cover the current bibliography on this topic. The authors need to improve this aspect along the whole manuscript.
-I did not find Figure 2 very informative, I suggest to remove it. I would rather design a figure related to the section entitled “RUNX modulates p53-dependent cell death”. Indeed, this section is quite dense and refer to many different actors. A figure summarizing this section would help to grab the main message.
-I suggest to change the title of the section “RUNX reduces DNA damaging stimuli” which appears quite confusing to me.
-In figure 1, I did not understand the meaning of the red star corresponding to “transcription”. The authors should better explain in the figure legend what they refer to here.
- Table 1 is missing and therefore could not be reviewed.
Reviewer 3 Report
This is a review of the role of the RUNX protein family in genome repair. It notes that the review strives to provide new insights into the RUNX family protein’s transcriptional and non-transcriptional regulation of DNA repair.
1) There are many issues with the wording and grammar throughout the article which make it difficult to easily read. In addition, certain strange phrases are used that are not typical for the meaning intended. Just for one example, on page 4 the authors state “tweaked to work…” It would help to have an English-speaking individual review this manuscript in detail to eliminate these issues.
2) A major issue is that it is unclear what this review is providing above what other recent review articles have already provided on this topic. For example, the review by Samarakkody, et al., in Mol. Cells 2020 on the Role of RUNX Family Transcription Factors in DNA Damage Response. Or the regulatory role of RUNX1, RUNX3 in the maintenance of Genomic Integrity by Krishnan and Ito, Adv Exp Med Biol, 2017.
3) The authors note on pg 1 that “Moreover, studies involving US soldiers and residents who were exposed to radiation from the above ground nuclear bomb test in Bikini atoll or Nevada from 1940s to 1950s developed MDS/AML due to RUNX1 point mutations or RUNX1 related chromosomal translocation”. The paper by Osato is cited. However, this paper clearly notes that RUNX1 abnormalities per se are insufficient to cause full-blown leukemia, yet this statement implies that the RUNX1 mutations are causing AML. The authors meaning should be clarified.
4) It is not clear why the heading “RUNX reduces DNA damaging stimuli” is used for a section noting the RUNX proteins regulation of ROS levels and oxidative stress. Wouldn’t the latter be a more apt title that than “DNA damaging stimuli”?
5) The figures also do not add much or provide much information, they are very basic and vague. In addition, the figures are blurry and not crisp, there is some background shading behind the letters, as if they were copies of copies.
6) In a few places it is implied that there is only senescence and apoptosis in terms of cell fate if DNA repair fails. However, there are other outcomes as well such as autophagy and mitotic cell death.
7) The conclusion is very brief and vague. There should be a more directed role for this review that is above what has been provided by others and in the conclusion some idea of future studies and what is still needed to flush out the role of this family of proteins in genomic stability and DNA repair. Without this the paper does not seem very useful.
Reviewer 4 Report
This review article is well written, well argued and comprehensive.
Round 2
Reviewer 2 Report
Figure 1 seems to be missing from the revised version of the manuscript.
The authors addressed several of my concerns. I still regret that they did not take the opportunity to include a figure illustrating the section entitled "RUNX modulates p53-dependent cell death"
